# The Influence of the Content of Recycled Natural Leather Residue Particles on the Properties of High-Density Fiberboards

**DOI:** 10.3390/ma16155340

**Published:** 2023-07-29

**Authors:** Katarzyna Bartoszuk, Grzegorz Kowaluk

**Affiliations:** 1Faculty of Wood Technology, Warsaw University of Life Sciences—SGGW, Nowoursynowska St. 159, 02-787 Warsaw, Poland; s207494@sggw.edu.pl; 2Institute of Wood Science and Furniture, Warsaw University of Life Sciences—SGGW, Nowoursynowska St. 159, 02-776 Warsaw, Poland

**Keywords:** fibers, natural leather, upholstery furniture, fiberboard, HDF, recycling

## Abstract

During the production of furniture, large amounts of waste materials are generated, which are most often stored in warehouses without a specific purpose for their subsequent use. In highly developed countries, as many as 25 million tons of textile waste are produced annually, of which approximately 40% is non-clothing waste such as carpets, furniture and car upholstery. The aim of this research was to produce and evaluate dry-formed high-density fiberboards (HDF) bonded with urea-formaldehyde resin, 12% resination, with various shares of recycled particles of natural leather used in upholstery furniture production at different contents (1, 5 and 10% by weight). The panels were hot-pressed (200 °C, 2.5 MPa, pressing factor 20 s mm^−1^). Mechanical properties (modulus of rupture, modulus of elasticity and screw withdrawal resistance) and physical properties (density profile, thickness swelling after water immersion, water absorption and surface absorption) were tested. The density profile and contact angle of natural leather have been also characterized. The results show that increasing the content of leather particles in HDF mostly has a positive effect on mechanical properties, especially screw withdrawal resistance and water absorption. It can be concluded that, depending on the further use of HDF, it is possible to use recovered upholstery leather particles as a reasonable addition to wood fibers in HDF technology.

## 1. Introduction

Wood-based panels are often used as an alternative to solid wood. Despite the content of veneers, fibers or wood particles, the boards sometimes have better properties and structure than natural raw material. HDF boards are fiber-based boards that are produced using the dry method. In addition, these boards are characterized by a very high density of over 800 kg m^−3^ [1]. Wood has always been one of the main sources of many materials, industries and raw materials for survival in the world. The growing development of pro-environmental movements and the pursuit of sustainable management of wood raw materials often leave a dilemma as to whether to use the available resources or look for new ways, materials and opportunities in the wood industry [2]. Unfortunately, in recent years, massive deforestation in some regions [3], constantly changing regulations, price increases, forecasting paper consumption increases [4] and reduced imports of wood from abroad [5] have meant that we have to replace wood raw materials with other available products. Such an alternative may be cereal straw used in the production of fiberboards [6]. In addition to natural raw materials such as wood or cereal straw, the furniture industry also uses huge amounts of natural and synthetic leather, fabrics and upholstery materials. During the production of furniture, large amounts of waste of these materials are generated, which are most often stored in warehouses without a specific purpose for their subsequent use. In highly developed countries, as many as 25 million tons of textile waste are produced annually, of which approximately 40% is non-clothing waste such as carpets, furniture and car upholstery [7]. It should be pointed out that Poland is one of the biggest producers of furniture in Europe; about 30.7% of total furniture production is upholstery furniture [8]. An example of its utilization is the mixing of Wet White and Wet Blue leather particles that have been mixed with spruce and beech wood fibers. Five different combinations with the amount of leather have been used, and boards have been produced by both the dry and the wet method. Research has shown, among other things, non-linear improvement of the internal binder with increasing leather content for HDF (4.5 mm, 900 kg m^−3^) [9]. There have also been trials to successfully utilize the textile dust in a share of 20% in the core layer of particleboards [10]. Some of the carbon-rich materials, including natural leather, can be thermally transformed into higher-value raw materials and products [11]. In another study [12], different content (0–20%) of the addition of fibers from upholstered fabric waste in HDF boards was used. The tests showed a slight decrease in the modulus of elasticity and the modulus of rupture with the increase in fiber content, but they were still within the limits given by the standard. On the other hand, there was no relationship between water absorption and thickness swelling. An example of the use of currently available natural resources for the production of panels is research that can be conducted in Hungary [13]. A total of 80% of poplar fibers and 20% of fibers from other available trees, e.g., black locust and Austrian pine, have been used. Urea-formaldehyde glue and ammonium sulphate-based hardeners were used for the production of the boards. Some paraffin has also been added to increase moisture resistance. Apart from the internal bonding values, the results were very satisfactory, well above the standard requirements. Some research has focused on the use of leather waste in wood-based composites, highlighting its potential as a reinforcing material. These review articles [14,15] present current research trends and prospects for the use of leather waste to produce composite polymers that are further transformed, for example into smart fertilizers. They produced value-added boards using leather particles and partially liquefied bamboo fiber. The mechanical and chemical performance capabilities were tested, followed by thermal insulation and sound absorption properties. When it comes to the addition of fibrous, recycled materials to wood-based composites, the regenerated cellulose [16] and chicken feathers [17] have been successfully tested. In general, according to [18], textile waste recycling is a growing and challenging task for the global economy.

This research aimed to produce and evaluate HDF with various shares of recycled particles of natural leather used in upholstery furniture production. The novelty of this research is the approach to recycling the raw materials from upholstery furniture production, providing the characterization of the produced particles and producing the HDF panels (boards), and, finally, evaluating the features of the produced panels in light of the raw materials used.

## 2. Materials and Methods

### 2.1. Materials

The raw materials listed below were used to make the test material: virgin pine (*Pinus sylvestris* L.) debarked round wood from Polish State Forests (Podlaskie voivodeship, Orla, Poland) was used to produce the fibers. Virgin fibers were produced on an industrial Metso Defibrator EVO56 (Metso, Helsinki, Finland) with a 2.5 m diameter disc with 10 knives. The moisture content of the fibers was 3.8%. Commercial urea-formaldehyde (UF) resin (Silekol Sp. z o.o., Kędzierzyn-Koźle, Poland) of about 66.5% of dry content [19] with a formaldehyde to urea (F:U) molar ratio of 0.89, pH of 9.6 and viscosity of 470 mPa s was used. The resination was set at 12% of dry resin calculated on dry fibers with 3.0% of ammonium nitrate hardener, both calculated regarding the dry resin content. The curing time of the adhesive mass, composed as mentioned above at 100 °C, was about 88 s. No further hydrophobic agents were added. The size of the upholstery natural leather particles used in the research and obtained by manual–mechanical shredding of the sheets was between 2 and 0.1 mm. The contact angle on the wood fibers and upholstery natural leather (left and right side) was measured with the use of distilled water using the PHOENIX 300 Goniometer (Surface Electro Optics Co., Ltd., Suwon, Republic of Korea) by 5 repetitions on every type of raw material.

### 2.2. Preparation of Panels

The test material was laboratory-made dry-formed fiberboards with an aimed density of 840 kg m^−3^, a width and length of 320 × 320 mm^−2^ and a nominal thickness of 3 mm, with 3 replicates per every panel type. The following variants of the panels were produced: reference panels and panels with various upholstery natural leather particles content (1, 5 and 10% w/w referred to board weight) added at the panel production stage. Reference boards were made without the addition of upholstery leather particles. The wood fibers were divided before resination into three layers—one inner (68% by weight) and two outer (2× 16%). The formulation of the manufactured panels has been presented in Table 1. Leather particles were added to the inner layer fibers only during the resination stage. The mats were formed manually. The board pressing parameters (hydraulic press AKE, Mariannelund, Sweden) were the temperature of 200 °C, the pressing factor of 20 s mm^−1^ of the nominal board thickness and the maximum unit pressing pressure of 2.5 MPa. After the production of the boards, they were stored at 20 °C and 65% humidity until constant weight was obtained. The pictures of the cross-section of the composites with natural leather particles are presented in Figure 1. The darker dots on the cross-sections indicate the leather particles’ presence.

### 2.3. Characterization of the Manufactured Panels

Subsequently, the following physical and mechanical properties were determined in accordance with European standards: density (EN 323) [20], screw withdrawal resistance (SWR) (EN 320) [21], bending strength (modulus of rupture—MOR), modulus of elasticity (MOE) (EN 310) [22], internal bonding (IB) (which was determined according to EN 319 [23]), water absorption (WA) and swelling thickness (TS) after 2 and 24 h of immersion (EN 317) [24]. The mechanical properties were tested on a computer-controlled universal testing machine delivered by the Research and Development Centre for Wood-Based Panels Sp. z o. o. (Czarna Woda, Poland). For each test of mechanical and physical parameters, no fewer than 8 samples of each type of panel were used. To determine the density profile (DP), test specimens cut into 50 mm × 50 mm dimensions were used and analyzed on a Grecon DA-X measuring instrument (Fagus-GreCon Greten GmbH and Co. KG, Alfeld/Hannover, Germany) with direct X-ray densitometry scanning across panel thickness in 0.02 mm increments. Three samples of each test variant were tested, but one representative density profile for each panel type was selected for further evaluation. The selected results, whenever applicable, were referred to as European standards [25].

### 2.4. Statistical Analysis

Analysis of variance (ANOVA) and t-tests calculations were used to test (α = 0.05) for significant differences between factors and levels, and, where appropriate, an IBM SPSS statistic base (IBM, SPSS 20, Armonk, NY, USA) was used. A comparison of the means was performed by the ANOVA test. The statistically significant differences in the achieved results are given in the Results and Discussion section whenever the data were evaluated. Where applicable, the mean values of the investigated features and the standard deviation indicated as error bars were presented on the plots.

## 3. Results and Discussion

### 3.1. Natural Leather Raw Material Characterization

The basic characteristics of the natural leather used in the research are presented in Figure 2. The average density and density profile are given in Figure 2a. As can be seen, the mean density of the leather sheet is about 540 kg m^−3^. That means that the density of leather is comparable to the density of the wood used for the production of fibers [26]. When analyzing the leather density profile, it can be seen that there is a zone of significantly higher density, about 825 kg m^−3^, which is called the “right side” (visible for upholstery furniture users), and there is also a zone with constant decreasing density, starting from about 600 kg m^−3^ for the thickness center to the “left side” of the sheet, invisible for the upholstery furniture user. The different densities and structures of the right and left sides of the leather sheet also influence the contact angle, presented in Figure 2b. As can be seen, the contact angle of the left side is higher than that of the right side. It can be caused due to the presence of the higher roughness, non-smooth and woolly surface of leather, which is significantly different from the right side, which is smooth. The in-time decreasing contact angle, as well as the surface roughness–contact angle relation can be confirmed by [27]. As it has been proven by Gumowska and Kowaluk [28], the contact angle can influence the physical properties of fiberboards.

### 3.2. Modulus of Rupture and Modulus of Elasticity

Figure 3 and Figure 4 show the dependence of the modulus of rupture and modulus of elasticity on leather particle content, respectively. It can be seen that the leather content does not drastically change the results. For the reference sample, the value of MOR is 41 N mm^−2^, and for 10% leather particle content it is 42.4 N mm^−2^. As for the MOE (Figure 4), the best result for MOR is shown by the sample with a 5% leather particle content (45.5 N mm^−2^). This slightly increasing tendency for MOR with the leather particles increase is the opposite of the results of Nemli et al. from 2019 [10], where the mechanical properties of particleboards decreased with textile dust content increase. The lowest MOE value occurs for the lowest natural leather particle content—3416 N mm^−2^ for leather particle content of 1%. The highest MOE (3863 N mm^−2^) is for the reference sample (0% content). On the other hand, if the samples with non-zero leather content are analyzed, it can be seen that the highest MOE (3804 N mm^−2^) is obtained in the sample with 5% leather particle content. Despite the various results and the lack of a specific relationship, the values of the modulus of rupture and modulus of elasticity still meet the requirements of the European standard (EN 622-5) [25]. It should be noted that the tested panels reach significantly higher MOR and MOE values when referred to [29,30], even if the density of the tested panels is about 50–90 kg m^−3^ lower than referred to in the mentioned literature. The statistically significant differences of mean MOR values have been found for reference panels when referred to the remaining ones, as well as for 5% panel when referred to the remaining ones. In the case of the MOE, the only statistically significant difference in mean values has been found between reference and 1% panels, and between reference and 10% panels.

### 3.3. Screw Withdrawal Resistance and Internal Bonding

The results of the measurement of internal bonding and screw withdrawal resistance are presented in Figure 5. In the case of screw withdrawal resistance (Figure 5a), there is a slightly increasing trend in the achieved average values, with increasing leather particle content. The increase between the value for boards with a leather particle content of 1 and 5% is 11 N mm^−1^ (less than 8%), while between a board with 5 and 10% leather particle content the difference is 5 N mm^−1^ (about 3%). For 10% leather particles content, the SWR is 156 N mm^−1^, and for 0% it is 141 N mm^−1^. There is no statistically significant difference between the average SWR values. The raising SWR with leather particle content increase is opposite to the tests, where the upholstery textile fibers have been added to the HDF structure [12]. A similar tendency of a slight increase of the tested value with the increasing amount of leather particle content has been found for internal bonding (Figure 5b). The pictures of the samples with leather particles after the IB test are presented in Figure 6. The remark of the positive effect of leather content on HDF IB can be confirmed by [9]. However, the values of standard deviation (error bars) can indicate that both factors, e.g., manual forming and leather particles/wood fibers blending, can influence the internal properties of the panels by an uneven distribution of the mentioned materials.

### 3.4. Density Profile

An examination of the density profile visualized in Figure 7 shows that, with higher leather particle content, the density in the outer layers of HDF boards decreases, while in the inner layers, it increases. The density value in the core layer increases to about 880 kg m^−3^ at 10% leather particle content, while for 0%, it is about 810 kg m^−3^. In surface layers, when the leather particle content increases to 5%, the density drops from about 940 kg m^−3^ to about 915 kg m^−3^. The highest difference in density has been found in the boards with 10% leather particle content, for which the average density profile is 840 kg m^−3^. As the proportion of leather particle fibers increases, the density difference between the face and core layers of the boards decreases. This remark is fully in line with the research on the incorporation of textile fibers into HDF panels [12]. The density distribution changes in the HDF face layers zone when raw material of different bulk density is added have been confirmed by Sala et al. [31]. This can have a significant impact on the quality of finishing layers on the boards, since, as mentioned by Henke et al. [32], the same density boards allow for different surface roughness to be reached, and, thus, different finishing quality. However, the density profile and the remaining parameters of the panels can be tuned by mat surface spraying before hot-pressing and by press temperature distribution [33].

### 3.5. Thickness Swelling, Water Absorption and Surface Water Absorption

The results of the measurement of thickness swelling and water absorption are displayed in Figure 8a,b, respectively. After 2 h of soaking, the intensity of the thickness swelling results is more pronounced for increasing leather particle content than for 24 h of soaking, where leather particle content has no significant effect. After 2 h, for 0% leather particle content, the thickness swelling value is 32.41, and for 10% content, it is 35.25%, an increase of 2.84%. After 24 h of soaking, for 0% content, the value of thickness swelling is 34.06%, and 10% content—35.82% (the difference was only 1.76%). It can be seen that the maximum thickness swelling of the tested boards, given by European standards [25], has been slightly exceeded. However, it should be raised, that in our study there was no hydrophobic agent used in the production of panels, and the resin used to resinate the fibers and particles did not lead to a water-resistant bonding line being achieved. It can also be concluded that the particle content of the leather in the boards does not have a significant influence on thickness swelling.

The results of water absorption of the tested panels of various content of natural leather particles are presented in Figure 8b. Due to the high scattering of the results, it is hard to evaluate the right values; however, it can be concluded that, after 2 h of soaking, the decrease in water absorption occurs with an increasing amount of leather content. This may be due to the higher hydrophobicity of leather particles, which has been proven by measuring the contact angle (Figure 2b). However, when evaluating the values of WA after 24 h of soaking, it can be said that the hydrophobic feature of the leather is less visible, and the differences between the WA of the samples of highest and lowest leather particles decrease.

The results of the surface water absorption of boards with different leather particle contents are shown in Figure 9. As can be seen, WA tends to remain at the same level. The highest results were obtained for panels with a 1% leather particle content of 4107 g m^−2^. Based on the results, it can be concluded that leather particles do not have a significant effect on surface water absorption.

## 4. Conclusions

The novelty of this research is the approach to utilizing non-wood waste from upholstery furniture production in wood-based composites such as HDF panels. The above work aimed to demonstrate the possibility of upcycling waste upholstery leather fibers by incorporating them into HDF boards. The results show that increasing the content of leather particles in HDF boards to 10% w/w has no significant negative effect on physical properties, including density profile, thickness swelling after immersion in water and absorption. Mechanical properties, particularly screw withdrawal resistance, have the greatest impact. Even the lowest values of the modulus of elasticity and modulus of rupture meet the requirements of European standards. It can be concluded that upholstery leather particles are a promising addition to fibrous-type boards, considering the subsequent use of the manufactured HDF. Such use can reduce the amount of upholstery leather waste and is a promising outcome in terms of the principles of a circular (closed-loop) economy, waste upcycling and carbon capture and storage (CCS) policies.

Further activity over the evaluation of wood-based composites with incorporated recycled materials that come from furniture industry waste can provide an assessment of the recycling potential of these materials.

## Figures and Tables

**Figure 1 materials-16-05340-f001:**
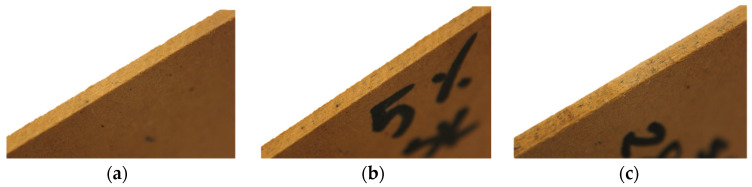
The pictures of the cross-cut of the tested panels of (**a**) 1% of natural leather particles, (**b**) 5% of natural leather particles and (**c**) 10% of natural leather particles (thickness about 3 mm).

**Figure 2 materials-16-05340-f002:**
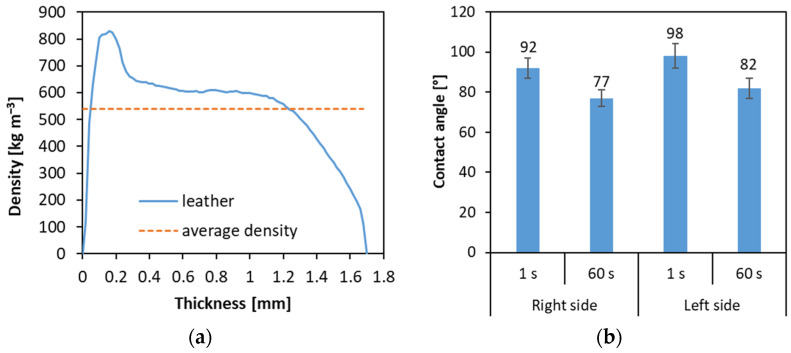
The average density and density profile (**a**) of natural leather sheets, as well as the contact angle of the leather surface (**b**).

**Figure 3 materials-16-05340-f003:**
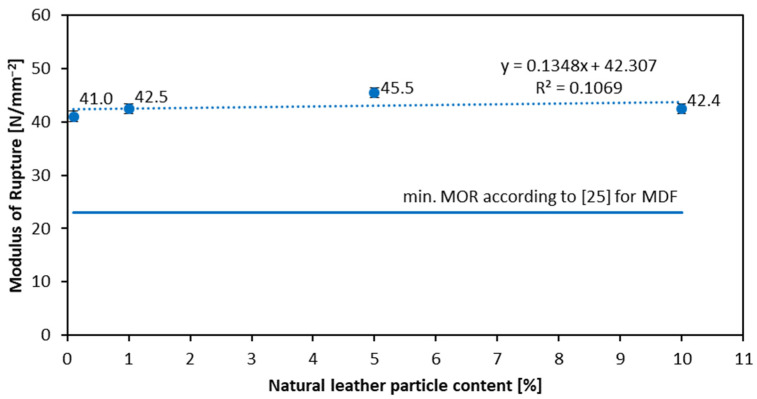
Influence of the natural leather particle content on the MOR of produced HDF.

**Figure 4 materials-16-05340-f004:**
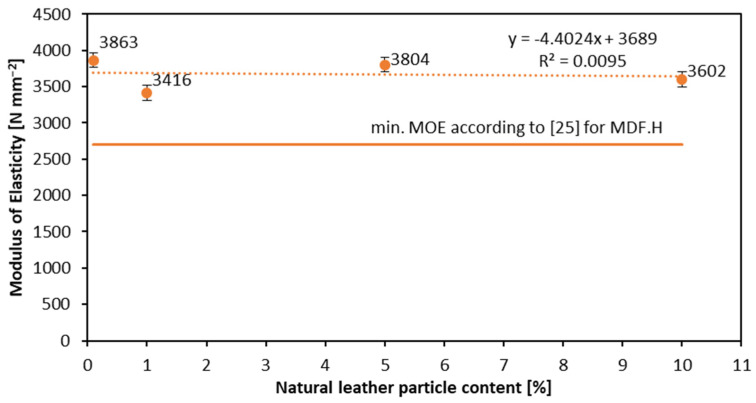
Influence of the natural leather particle content on the MOE of produced HDF.

**Figure 5 materials-16-05340-f005:**
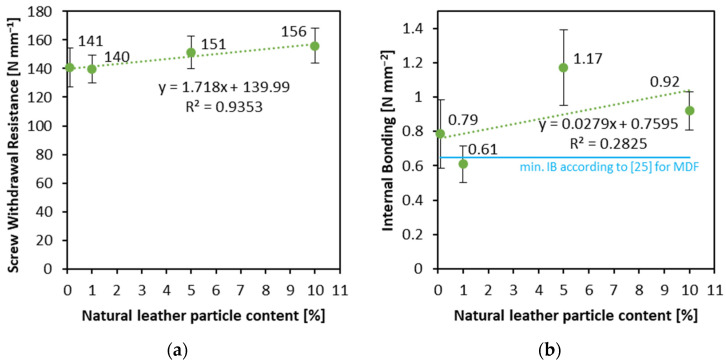
Screw withdrawal resistance (**a**) and internal bonding (**b**) of the panels with various content of natural leather particles.

**Figure 6 materials-16-05340-f006:**
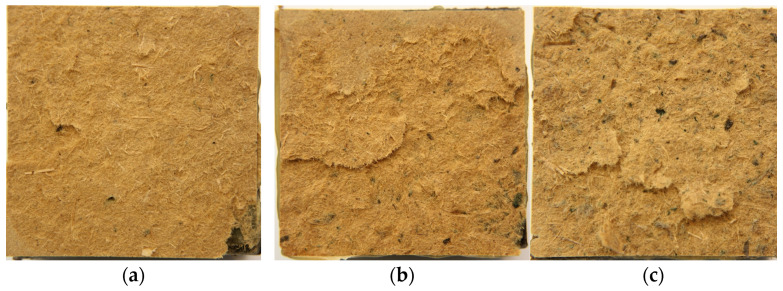
The pictures of the core layer of the tested panels of (**a**) 1% of natural leather particles, (**b**) 5% of natural leather particles and (**c**) 10% of natural leather particles after IB testing.

**Figure 7 materials-16-05340-f007:**
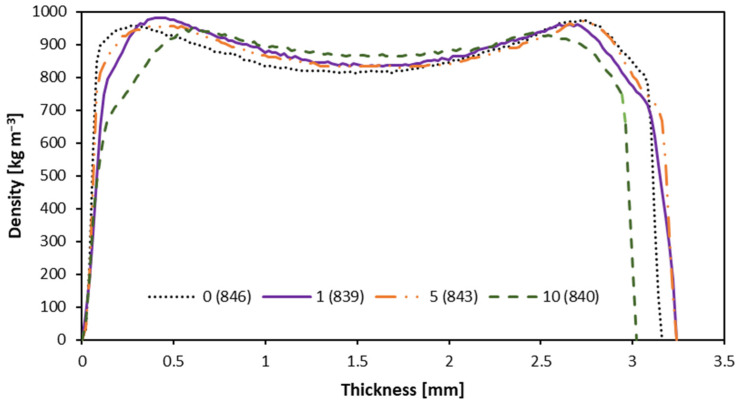
Density profiles of the panels produced by the use of different amounts of leather particles.

**Figure 8 materials-16-05340-f008:**
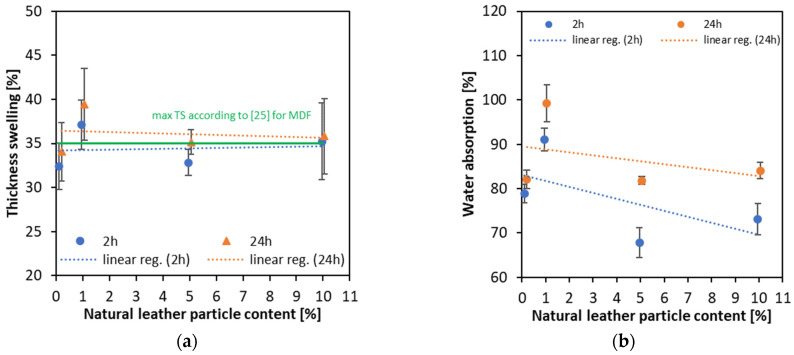
The thickness swelling (**a**) and water absorption (**b**) of the panels produced with the use of different amounts of natural leather particles.

**Figure 9 materials-16-05340-f009:**
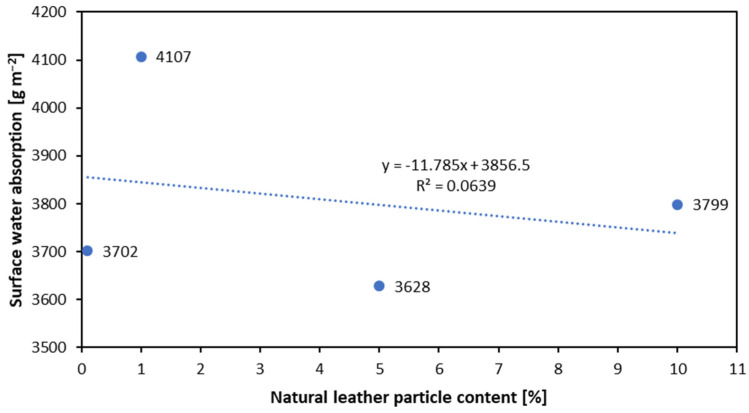
The surface water absorption of the panels produced with the use of different amounts of natural leather particles.

**Table 1 materials-16-05340-t001:** The general formulation of manufactured panels of various content of leather particles.

Panel Code	Natura Leather Particle Content in the Panel [%] ^1,2^	Face Layers Wood Fiber Content [g]	Core Layer Leather Particle/Wood Fiber Content [g]	Resination [%]
0% ^1^/reference	0	72.2	0.0/153.4	12
1%	1	72.2	2.3/151.2	12
5%	5	72.2	11.3/142.1	12
10%	10	72.2	22.6/130.9	12

^1^ by weight; ^2^ referred to total panel weight.

## Data Availability

The data presented in this study are available on request from the corresponding author.

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
