# Peer review of "The Influence of the Content of Recycled Natural Leather Residue Particles on the Properties of High-Density Fiberboards"

_materials, 2023, doi:10.3390/ma16155340_

Round 1

Reviewer 1 Report

The presented manuscript describes the properties of the resulting composite materials based on upholstery leather particles and wood fibers. The authors show that the introduction of leather particles into the matrix of wood fibers up to 10% does not impair the properties of the resulting materials. Here I would like to see a more detailed explanation of why this happens ?! For the graphs shown, for example 7, you can reduce the scale on the y-axis (3000-4500 g m-2). It is interesting to know the opinion of the authors about the future fate of such composite materials. If materials based on wood fibers can be recycled, then for the materials described, it is not entirely clear what to do next. It would be good to provide photographs of the distribution of components (optical or electron microscopy).

I recommend removing HDF from the list of keywords

Line 32. "both in Poland and" - I recommend deleting it.

Lines 35-38. I do not agree with the authors! According to a number of sources in Europe, on the contrary, the number of forests is increasing. As for the change in prices, it is rather due to other reasons (sone may ask why the price of wood remains high, if the prices for all types of fuel on the world market have long since dropped to the values that they were a few years ago?!). Also, one cannot leave aside the fact of a decrease in demand for a number of wood products, for example, newsprint, etc.

Line 56. "0, 5, 10, 20%)" I propose to replace it with "0-20%)".

Author Response

Dear Reviewer,
thank you for your opinion and contribution to improving our manuscript.
Attached, please find our responses to your remarks. The sufficient changes have been added to the manuscript, along with your comments.
With regards
Grzegorz Kowaluk

Reviewer 2 Report

This paper presents an interesting way of using waste upholstery leather generated during the production of upholstered furniture as an additive for HDF panels. This type of waste is usually stored because there is no idea how to use it. In proposing a way to use them, the authors be in line with current trends concerning the principles of a closed-loop economy, upcycling waste, and carbon capture and storage (CCS) policies.
The authors have approached the subject properly, conducting the research according to correctly chosen research methods.
My comments relate to the following points:
Please provide the characteristics of the UF resin - viscosity, gel time, and miscibility with water,...
What does the molar ratio of 0.89 mean? Urea to formaldehyde ratio or formaldehyde to urea ratio?
Please unify the naming of the boards or panels.
I doubt the correctness of the title "Characteristics of elaborated panels". Perhaps it would be better to change to "manufactured boards"? 

Author Response

(The authors gave the same response as above.)

Reviewer 3 Report

The study titled "The Influence of the Content of Recycled Natural Leather Residue Particles on the Properties of the HDF Boards" presented interesting results. However, authors should do a major revisions to clarify the results.

Please find the comments below:

Title : Please revise the title to “The Influence of the Content of Recycled Natural Leather Residue Particles on the Properties of the High-Density Fiberboards”.

Abstract:

·       Line 14-16. Please add the adhesive used, adhesive content in HDF, and the hot-pressing condition of HDF.

Introduction:

·       Line 29. Please write the full name of HDF and then follow by it’s short-name.

·       Line 30. What do you mean with HDF boards are fiber-based fibreboards? Please revise to “HDF boards are fiber-based boards”.

·       Line 52. Revise high-density boards to “HDF”.

·       Line 77. Just write “HDF” instead it’s full name.

Materials and Methods:

·       Line 92. Why did authors used ammonium nitrate hardener, while in introduction they mentioned ammonium sulphate. Please explain the reason.

·       Line 101. Did you prepare MDF or HDF? Please write it consistently.

·       Line 111. Please write the pressing time.

·       Please prepare a Table consists of a Formulation of HDF panel at different contents of Recycled Natural Leather Residue Particles.

·       Line 118. Please revise to “internal bonding (IB)”.

·       Please add the formaldehyde emission (FE) measurement of HDF, because the authors used UF resins as an HDF adhesive.

Results and Discussion:

·       Figures 2 and 3. The authors produced HDF but compared the results with MDF. This is wrong. MDF is medium-density while HDF is high-density.  Please revise it.

·       Please add a fair comparison between HDF and HDF

·       Similar to Figures 4 and 6. Please revise and put a fair comparison of HDF and HDF, not HDF to MDF.

Conclusions:

·       The conclusion is well written.

References:

·       The authors should add recent articles (last 5 years) as references.

Author Response

(The authors gave the same response as above.)

Round 2

Reviewer 3 Report

The revised manuscript is now better.

However the author should add the comparison of HDF properties from other published works.

The manuscript can be accepted after the minor revisions.

Author Response

Dear Reviewer,
according to your suggestion, as many as 5 more citations have been added to the manuscript. The mechanical properties (MOR and MOE), as well as the density profile, have been referred to the cited research.
Best regards!